# Ex-Vivo Pharmacological Defatting of the Liver: A Review

**DOI:** 10.3390/jcm10061253

**Published:** 2021-03-18

**Authors:** Claire Goumard, Célia Turco, Mehdi Sakka, Lynda Aoudjehane, Philippe Lesnik, Eric Savier, Filomena Conti, Olivier Scatton

**Affiliations:** 1Department of Hepatobiliary Surgery and Liver Transplantation, Sorbonne Université, Hôpital Pitié-Salpêtrière, Assistance Publique-Hopitaux de Paris, 75013 Paris, France; celia.turco@aphp.fr (C.T.); eric.savier@aphp.fr (E.S.); olivier.scatton@aphp.fr (O.S.); 2Sorbonne Université, Centre de Recherche Saint Antoine, INSERM UMRS-938, Institute of Cardiometabolism and Nutrition (ICAN), 75013 Paris, France; lynda.aoudjehane@inserm.fr (L.A.); filomena.conti@aphp.fr (F.C.); 3Department of Metabolic Biochemistry, Sorbonne Université, Hôpital Pitié-Salpêtrière, Assistance Publique- Hopitaux de Paris, 75013 Paris, France; mehdi.sakka@aphp.fr; 4Sorbonne Université, INSERM UMRS-1166, Institute of Cardiometabolism and Nutrition (ICAN), 75013 Paris, France; philippe.lesnik@sorbonne-universite.fr

**Keywords:** normothermic machine perfusion, ex-vivo perfusion, liver perfusion, steatosis, defatting, liver transplantation

## Abstract

The ongoing organ shortage has forced transplant teams to develop alternate sources of liver grafts. In this setting, ex-situ machine perfusion has rapidly developed as a promising tool to assess viability and improve the function of organs from extended criteria donors, including fatty liver grafts. In particular, normothermic machine perfusion represents a powerful tool to test a liver in full 37 °C metabolism and add pharmacological corrections whenever needed. In this context, many pharmacological agents and therapeutics have been tested to induce liver defatting on normothermic machine perfusion with promising results even on human organs. This systematic review makes a comprehensive synthesis on existing pharmacological therapies for liver defatting, with special focus on normothermic liver machine perfusion as an experimental ex-vivo translational model.

## 1. Introduction

Liver steatosis (LS) and its consequences, known as non-alcoholic fatty liver disease (NAFLD), have taken place in the top three indications for liver transplantation in Western countries, owing to the growing epidemics of obesity and metabolic syndrome [1,2]. In addition to representing a global health issue, liver steatosis is also a limiting factor for its own treatment: liver transplantation (LT). The proportion of organ donors with obesity is rising, and an increasing number of potential liver grafts are discarded due to high steatosis. Although prevention for obesity may help limiting LS, preventive strategies are not sufficient to overcome the health issues related to this condition [3,4]. 

In this context, research on liver defatting therapies may represent a promising perspective. From In Vitro models to ex-vivo human liver perfusion, several pharmacologic substances have already been evaluated and are currently under investigation [5]. 

In parallel, non-pharmacological strategies, such as ex-vivo liver machine perfusion, have been applied on highly steatotic liver grafts, with promising results on organ function improvement but no impact on organ fat content. However, discarded highly steatotic grafts, which are not suitable for transplantation but can be maintained on normothermic machine perfusion (NMP), may represent the ultimate experimental model to assess various therapeutic defatting strategies. Moreover, the direct impact on the possible use of these grafts for transplantation, if the defatting succeeds, constitutes a goal in itself for developing such therapies [6]. 

This systematic review aims at making a comprehensive synthesis on existing pharmacological therapies for liver defatting, with special focus on normothermic liver machine perfusion as an experimental ex-vivo translational model. 

## 2. Materials and Methods

This systematic review has been conducted according to the Preferred Reporting Systematic Reviews and Meta-Analyses (PRISMA) guidelines [7]. 

Papers from the following electronic databases were searched until 1 December 2020: PubMed, Web of Science, EMBASE, MEDLINE, Cochrane Library and Scopus. The medical subject heading (MeSH) terms and keywords were “liver steatosis”, “fatty livers”, “liver defatting” and “steatosis reversal”.

Inclusion criteria were articles assessing defatting therapies, including animal models or human liver experimentation and in-vitro models; and articles written in English. 

Two authors (C.G. and C.T.) selected the articles independently for inclusion in the review in three steps. The first step reviewed all article titles to exclude those not matching inclusion criteria, then the remaining abstracts were analyzed in the second step. Abstracts considered as not relevant were excluded. Finally, remaining full papers were analyzed and included in the review. 

## 3. Results

The first step of searching retrieved 227 (“liver defatting”) results from which 20 relevant abstracts were included. After exclusion of studies out of the scope of the present review, 13 full text articles were included (Figure 1).

### 3.1. Steatosis Physiopathology and Its Consequences 

Liver steatosis is defined as the accumulation of triglycerides (TG) within the cytoplasm of hepatocytes. The size of the lipid droplets determines their classification in macrovesicular steatosis (large droplets which displace the nucleus) or microvesicular steatosis (small droplets). While the clinical repercussions according to the type of steatosis is still debated, it is accepted that a macrosteatosis over 60% is associated to liver damage and dysfunction [8]. As a consequence, those livers are usually declined for transplantation. 

The damage caused by steatosis comes from different mechanisms: (1) impaired cellular metabolism, with difficulty to stock ATP, due to altered mitochondrial activity. The impaired ATP synthesis and stockage activity leads to extensive cellular injury when those liver are exposed to stress such as ischemia-reperfusion; (2) microvascular obstruction due to mechanic reduction of the sinusoids diameter; (3) pro-inflammatory environment caused by activation of Kupffer cells with overproduction of reactive oxygen species (ROS) [5,9].

The combination of those structural weaknesses in steatotic livers leads to a reduced capacity to endure ischemia. Liver grafts are usually stored under static cold conditions (ideally 4 °C) for several hours (up to 12 h, ideally 8 h maximum) until they undergo reperfusion within the recipient. The inability of restocking ATP and the higher production of ROS in association to the altered microcirculation obstructing correct perfusion lead to high vulnerability to ischemia/reperfusion, which has been correlated to poor graft outcome [10,11]. Transplanted livers with >60% steatosis have been associated with high rates of primary non function (PNF) and increased morbidity after transplantation [12,13]. 

Steatosis has been summarized as “the result of an imbalance between TG synthesis and breakdown process in hepatocytes” [1]. One way to overcome this issue would be to shift this balance towards more efficient TG breakdown (lipolysis) and excretion of related bioproducts along with TG synthesis reduction. To reduce the amount of intracellular lipids, which are stored as triglycerides, two major pathways can be targeted: (1) secretion of TG through very low-density lipoprotein (VLDL), and (2) fatty acid oxidation. In both cases, the stored triglycerides first undergo lipolysis then either re-esterification and assembly with apolipoproteins (such as apolipoprotein B-100) into TG-rich VLDL, or complete hydrolysis into glycerol and fatty acids [9,14]. Figure 2 shows the main pathways for liver defatting and the molecules used in the literature to trigger defatting. 

### 3.2. Pharmacological Therapies for Liver Defatting: In Vitro and Animal Models

#### 3.2.1. Animal Hepatocyte Models 

In Vitro defatting models are summarized in Table 1. Hepatocyte culture models demonstrated the reversibility of liver steatosis, with direct implications on the hepatic sensitivity to IRI [8]. Berthiaume et al. showed cultured steatotic rat hepatocytes generated more mitochondrial superoxide, exhibited a lowered mitochondrial membrane potential and released significantly more lactate dehydrogenase after hypoxia and reoxygenation than lean hepatocyte controls. When steatotic hepatocytes were defatted by incubating in fatty acid-free medium, they became less sensitive to hypoxia and reoxygenation as the remaining intracellular triglyceride content decreased. In a Lewis rat LT model, transplanted steatotic livers had 0% viability compared with 90% for lean liver controls. When donor choline and methionine-deficient diet rats were returned to a normal diet, hepatic fat content decreased while viability of the grafts after transplantation increased [8].

It was reported in 2004 that treating 48 h ob/ob mice with the fatty acid synthase inhibitor cerulenin induced a shift from macro to microsteatosis and was associated with improved survival after I/R injury. Moreover, treatment with cerulenin for 2, 4 or 7 days before liver procurement and transplantation increased recipient survival proportionally [15].

The first published defatting pharmacological cocktail was developed by Pégorier et al. in 1989 [16]. Hepatocytes from fetal and newborn rabbits were cultivated during 4 days in the presence of glucagon, forskolin, dibutyryl cyclic AMP, 8-bromo cyclic AMP or insulin. The authors showed glucagon, forskolin and cyclic AMP induced ketogenesis and fatty acids oxidation through changes in the partitioning of long-chain fatty acid from esterification towards oxidation [16]. These results were confirmed by the same team in human hepatocytes (developed in Section 3.2.2 on human hepatocytes model) [17]. 

The concept of liver defatting cocktail was adapted more recently by Nagrath et al. and confirmed in the setting of rat liver machine perfusion [18]. Their pharmacological cocktail included the five following components: (1) PPARδ ligand **GW501516 (GW5)** which stimulate transcription of lipid oxidation and export factors, (2) Pregnane X receptor (PXR) ligand **Hypericin (HSC)**, which improves transcription of pro-β-oxidation cytochrome P450(CYP)34 Amono oxygenase, (3) Constitutive androstane receptor (CAR) ligand **Scoparone (SCO)**, which promotes transcription of β-oxydation enzymes (carnitine palmytoltransferase), (4) Glucagon mimetic and cAMP activator **Forskolin (FOR)**, which stimulates cyclic cAMP-driven β-oxydation and ketogenesis, (5) insulin mimetic adipokine **Visfatin (VIS)**, which decreases triglyceride levels in liver (Figure 2). Nagrath’s defatting cocktail first results on hepatocyte rat culture induced the reduction of intracellular lipid content (31% reduction over 48 h) and promoted triglyceride export (increase of ketone bodies production by 24%, mitochondrial O_2_ uptake by 1.8-fold and TG secretion by 1.4-fold), as well as each DFAT agent individually. 

Later in 2014, Nativ et al. published their own defatting cocktail, which included the same components as Nagrath’s cocktail with **L-Carnitin** addition and hyperoxic perfusion conditions (90% O_2_) [19]. L-Carnitin is a transporter of fatty acids across the inner mitochondrial membrane. This modified cocktail, resulted in an 82% decrease of lipid droplets in primary macrosteatotic hepatocyte cultures from male Zucker rats, along with TG content reaching a maximal decrease when L-carnitin was added in hyperoxic conditions (original cocktail 27% TG content increase, +L-Carnitin 41%, +hyper Ox 45%, +L-Carnitin and hyperOx 57%).

Last, the defatting effect of RIPA-56, a highly specific inhibitor of RIPK1, which is a gatekeeper of the necroptosis pathway, was recently demonstrated by Majdi et al. [20]. Necroptosis is a regulated form of necrotic cell death mediated by the receptor-interacting protein kinase (RIPK) 1 that is activated in NAFLD. When used as either a prophylactic or curative treatment for HFD-fed mice, RIPA-56 caused a downregulation of MLKL and a reduction of liver injury, inflammation and fibrosis, as well as of steatosis. 

#### 3.2.2. Human In Vitro Models

In Vitro defatting models, including human hepatocytes models, are summarized in Table 1. Two main types of In Vitro human hepatocytes models are used in the literature, either HEPG2 cells obtained from primary liver cancer derived lines, or hepatocytes isolated from liver biopsies. In Vitro hepatocyte model isolated from human livers carry the advantage of expressing enzymes involved in drug metabolism such as CYP450, in contrast to HEPG2 cells or rat hepatocytes where this expression may be more variable. 

The first published defatting pharmacological cocktail developed by Pégorier et al. in 1989 [16] confirmed their results in human hepatocytes cultured from liver biopsy specimens in an insulin-free medium, where glucagon or cyclic AMP significantly enhanced ketone body production and oleate oxidation [17]. 

The modified cocktail from Nativ et al., which included the same components as Nagrath’s cocktail with **L-Carnitin** addition [19], was further evaluated by Yarmush et al. on steatotic Human Hepatoma (HepG2) Cells [21], who reported a 50% intracellular TG decrease after 48 h exposure to the cocktail, this decrease occurring within the first 24 h. Hyperoxic conditions (90% O_2_ vs. 21%) were associated with lower TG levels. The same author further demonstrated that flow mediated transport of metabolites reduced defatting kinetics from 48 to 4–6 h in the same HepG2 culture model [22].

Boteon et al. published their own modified cocktail which consisted of Nagrath’s cocktail with PPARα ligand and L-Carnitine addition [23]. Its incubation of 48 h with steatotic human hepatocytes isolated from discarded donor livers showed a 32% decrease in intracellular TG content and a 54% decrease in lipid droplets quantity, which gained the appearance of microvesicular steatotic droplets.

More recently, another defatting cocktail was proposed by Aoudjehane et al. which included Forskolin, L-carnitine and a PPARα combined with Rapamycin, an immunosuppressive drug that induces autophagy [24]. The cocktail was tested alone or in combination with necrosulfonamide, an inhibitor of mixed lineage kinase domain like pseudokinase involved in necroptosis. Three human culture models were used: primary hepatocytes with induced steatosis, primary hepatocytes isolated from steatotic liver and precision-cut liver slices (PCLS) of steatotic liver. In all three models, D-FAT induced a 30% decrease in TG content within 24 h along with a decrease in endoplasmic reticulum stress and reactive oxygen species production. The addition of necrosulfonamide increased the efficacy of defatting by 8–12% in PCLS [24]. 

The defatting effect of RIPA-56 shown on mice livers by Majdi et al. [20] was confirmed by treating primary human steatotic hepatocytes with RIPA-56 vs. necrosulfonamide, a specific inhibitor of human MLKL. Mlkl-KO led to activation of mitochondrial respiration and an increase in β-oxidation in steatotic hepatocytes. The defatting effect of RIPA-56, in addition to its role on liver injury and inflammation reduction, may be interesting to analyze in future human NMP studies, either alone or in combination to other defatting molecules.
jcm-10-01253-t001_Table 1Table 1Drugs used in in vitro models for defatting of steatotic hepatocytes.
Model*n*Defatting AgentsMain OutcomesPégorier et al.,1989 [16]rat hepatocytes8glucagon, forskolinand c-AMPinduction of ketogenesisBerthiaume et al.,2009 [8]rat hepatocytes14Nonesteatosis reversalNagrath et al., 2009 [18]Rat hepatocytes4combination of visfatin, forskolin, hypericin and nuclear receptor ligands (GW7, GW5, scoparone)reduction of intracellular TGcontentpromotion of lipid exportNativ et al.,2014 [19]rat hepatocytes6defatting cocktailreduction of intrahepatic TGreduction of large lipid dropletsYarmush et al.,2015 [21]HepG2 cells3combination of visfatin, forskolin, hypericin and nuclear receptor ligands (GW7, GW5, scoparone)decrease of TG contentBoteon et al.,2018 [23]human hepatocytes from discarded donor liversHIEC, cholangiocytes4combination of visfatin, forskolin, hypericin, L-carnithine, PPARα ligand and nuclear receptor ligands (GW7, GW5, scoparone)reduction of intracellular TGinduction of fatty acids β-oxidationAoudjehane et al.,2020 [24]human hepatocytes from fatty livers6forskolin, L-carnitine and PPARα agonistreduction of intracellular TGMadji et al.,2020 [20]mice livershuman steatotic hepatocytes1010RIPA-56decrease of intracelliular lipid droplets and TG contentdecreased inflammation and liver injuryPégorier et al.,1989 [16]rat hepatocytes8glucagon, forskolinand c-AMPinduction of ketogenesisBerthiaume et al.,2009 [8]rat hepatocytes14NAreversing of steatosisNativ et al.,2014 [19]rat hepatocytes6defatting cocktailreduction of intrahepatic TGreduction of large lipid dropletsYarmush et al.,2015 [21]HepG2 cells3combination of visfatin, forskolin, hypericin and nuclear receptor ligands (GW7, GW5, scoparone)decrease of TG contentBoteon et al.,2018 [23]human hepatocytes from discarded donor liversHIEC, cholangiocytes4combination of visfatin, forskolin, hypericin, L-carnithine, PPARα ligand and nuclear receptor ligands (GW7, GW5, scoparone)reduction of intracellular TGinduction of fatty acids β-oxidationAoudjehane et al.,2020 [24]human hepatocytes from fatty livers6forskolin, L-carnitine and PPARα agonistreduction of intracellular TGMadji et al.,2020 [20]mice livershuman steatotic hepatocytes1010RIPA-56decrease of intracelliular lipid droplets and TG contentdecreased inflammation and liver injuryAMP: cyclic adenosine monophosphate; TG: triglycerides; HIEC: human intra-hepatic endothelial cells.

#### 3.2.3. Animal NMP Models

Published experiments for ex-vivo defatting of steatotic livers are summarized in Table 2. Following first encouraging results on rat hepatocytes, Nagrath’s cocktail was administered in fatty livers isolated from obese Zucker rats and maintained 3 hours on normothermic ex-vivo machine perfusion. The TG content was reduced by 65% compared to 30% in control, and TG secretion was increased by 40% compared to control with a two-fold increase in ketone bodies secretion. Thus, Nagrath’s DFAT cocktail demonstrated an effect on both two major pathways for reducing intracellular lipid content: secretion through VLDL (export), and fatty acid β-oxidation. Interestingly, perfused organs showed more rapid defatting than cultured cells, possibly due to the presence of non-parenchymal cells and the possibility of full organ metabolism.

Those results were challenged by Liu et al. who tested Nagrath’s DFAT cocktail in Zucker rat livers perfused 6 h at room temperature, corresponding to subnormothermic conditions [25]. They found increased VLDL and TG content in perfusate with or without defatting cocktail as well as active lipid metabolism export and no consistent changes in histology. These results may be explained by the subnormothermic conditions which limit active metabolism and oxygen availability, therefore restricting full β-oxidation and lipid active export. 

Nativ’s modified cocktail, which added L-Carnitin to Nagrath’s one, was further evaluated by Raigani et al. on steatotic rat liver under 6 h normothermic hyperoxic machine perfusion [26]. Defatting cocktail perfusion was associated with decreased pro-inflammatory signaling (NFkB, TNFα), increased hepatic gene expression of mitochondrial fatty acid β-oxidation markers and moderately reduced tissue TG. Macrosteatosis decreased during perfusion of steatotic livers with or without defatting cocktail. However, defatting perfusion was associated with higher transamisase levels, significantly depleted ATP production and increased insulin resistance, suggesting some degree of hepatotoxicity and energy depletion. 

Last, Taba Taba et al. investigated the defatting effect of Glial Cell Line-Derived Neurotrophic Factor (GDNF) in mice livers under 4 h normothermic perfusion [27]. GDNF induces expression of PPARα, increases lipolysis and enhances adrenergic mediated cAMP release. GDNF and defatting cocktail were equally effective in steatotic liver defatting, with less LDH activity with GDNF, interpreted by the authors as reduced liver tissue damage. 

### 3.3. Defatting and Ex-Vivo Human Liver Machine Perfusion

The emergence of liver machine perfusion has allowed to push forward research on liver metabolism and offers a unique platform to evaluate therapeutics that may improve liver grafts function [6,28,29]. Published experiments for ex-vivo defatting of steatotic livers are summarized in Table 2.

The defatting effect of normothermic liver perfusion in itself was investigated prior to any pharmacologic addition. Indeed, the full 37 °C perfusion may constitute a metabolic stress and induce a certain level of lipolysis. In a porcine model, Jameson et al. investigated 48 h continuous NMP of fatty livers, demonstrating a histologic decrease in hepatic fat content from 30% to 15% [30]. 

However, Banan et al. did not confirm these results on perfused human livers [31]. They perfused 7 discarded human livers on a modified normothermic circuit during 8 h. Two of the 7 organs were steatotic and underwent addition of a defatting solution comprising Exendin-4, a member of the glucagon-like peptide family which has been associated to the reduction of oxidative stress in steatotic models, and L-Carnitin. One steatotic liver did not undergo defatting cocktail perfusion and served as control. Defatted livers had increased perfusate levels of TG and LDL and at the end of experiments, (TG concentration 8.8 times higher at hour 8 than at hour 0) and a 10% reduction in histological steatosis. In the control liver, TG levels did not change, and LDL concentration decreased with time while steatosis amount remained unchanged on histology. These results support a specific effect of defatting molecules on lipid metabolism distinct from the global activating effect of normothermic perfusion on metabolism. 

Those results on human perfused livers were confirmed by Boteon et al. on 10 discarded livers which underwent NMP with the Liver Assist^®^ device, 5 with Nagrath’s DFAT cocktail injection, and 5 control [32]. The livers treated with DFAT showed a tissue TG decrease of 38% vs. 7% in controls over 6 h, while perfusate TG increased. Tissue ketogenesis and ATP levels increased while levels of pro-inflammatory cytokines TNFα and IL1β and lactate decreased. Treated DFAT livers also had a 40% reduction of macrovesicular steatosis on histology. Of note, all livers perfused did not display significant histological steatosis (one had none, and one severe steatosis per group). 

Although only two studies so far published the effects of a defatting cocktail injection on human livers during NMP, these preliminary reports showed promising results on lipid metabolism activation. While only representing data from 7 treated livers and 6 controls, these results need confirmation at a wider scale. Several questions remain, such as the treatment of perfusate full of exported TG; perfusion teams surely have to add supplementary membranes to their circuit to filtrate the fat-loaded perfusate. Moreover, the effect of fat content reduction on liver function improvement still remains to be demonstrated through larger experimental studies focusing on surrogates of liver metabolism and transplantability criteria.
jcm-10-01253-t002_Table 2Table 2Drugs used in experimental models for ex-vivo defatting of steatotic livers.
Model*n*PerfusionLength of Perfusion (hours)Defatting AgentsMain OutcomesNagrath et al., 2009 [18]rat7NMP3combination of visfatin, forskolin, hypericin and nuclear receptor ligands (GW7, GW5, scoparone)decrease of TG rateimprovement of bile productionLiu et al.,2013 [25]ratNASNMP6combination of amino acids, visfatin, forskolin, hypericin and nuclear receptor ligands (GW7, GW5, scoparone)higher rate of TG (non significance)Raigani et al., 2020 [26]rat6NMP6combination of amino acids, visfatin, forskolin, hypericin and nuclear receptor ligands (GW7, GW5, scoparone)decrease of pro inflammatory markers (NF- κB, TNF-α, IL-6)decrease of pro-apoptotic markers (CASP3, CD95)decrease of combined VLDL/LDL levelTaba Taba Vakili et al., 2016 [27]mice4NMP4glial cellline–derived neurotrophic factorreduction of TG content in liverno liver damage (no increase of apoptosis)Banan et al.,2016 [31]human discarded livers2NMP8L-carnithine and exendin-4decrease of TG and LDL level in the perfusatereduction of MasBoteon et al.,2019 [32]human livers discarded5NMP12combination of L-carnithine, visfatin, forskolin, hypericin and nuclear receptor ligands (GW7, GW5, scoparone)decrease of T-TG leveldecrease of MaSincrease of P-TG level higher bile productionLDL: low-density lipoprotein; MaS: macrovesicular steatosis; NMP: normothermic perfusion; P-TG: SNMP: subnormothermic perfusion; TG: triglycerides; T:TG: tissue-TG; VLDL: very low-density lipoprotein.

### 3.4. Future Perspectives

As human liver on NMP constitutes the optimal preclinical model, the development of various therapeutics with the aim of defatting remains widely open. As an example, Goldaracena et al. published one of the first large-animal studies to provide proof of concept for optimization and modification of liver grafts during NMP. They administered an antisense miRNA-122 oligonucleotide, which is required for hepatitis C replication, during NMP of porcine liver grafts prior to transplantation. They found suppression of HCV replication after established infection and prevented HCV infection with pretreatment of cells, analogous to the pretreatment of grafts in the transplant setting [33]. Use of the RNA interference (RNAi) pathway to silence specific genes implicated in IRI is one of the most recent and promising advances in the field of NMP liver therapeutics. The overall RNAi mechanism contains several regulatory RNA molecules including microRNA (miRNA), small interfering RNA (siRNA), and short-hairpin RNA (shRNA). So far, RNAi have been evaluated in IRI gene silencing with promising preliminary results in animal models [34]. Similarly, defatting therapies may take the form of miRNA or siRNA targeting lipolysis and lipid export genes. 

In parallel, the technology of NMP is rapidly developing, and a recent study reported a 7-days liver perfusion on a high technology device providing a complete organ artificial support [35]. However, although such technological achievements remain very exciting, their cost and logistics may be hard to export to many transplant centers. Another team recently reported the development of a highly adaptable circuit that can fit on already pre-existing extracorporeal oxygenation machines routinely used in cardiovascular surgery. This circuit would allow any type of perfusion conditions without interrupting the perfusion process and may constitute a simpler alternative to existing perfusion systems, while allowing easy additions of extra components such as filtration systems [36].

In conclusion, NMP is a promising platform for assessing novel therapeutics on fatty livers, with the goal of reducing fat content and in the end optimizing fatty liver grafts for transplant. Novel pharmacological defatting therapies already demonstrated encouraging preliminary results which require further validation. The field of research on liver grafts optimization carries wide and exciting perspectives in the forthcoming years. 

## Figures and Tables

**Figure 1 jcm-10-01253-f001:**
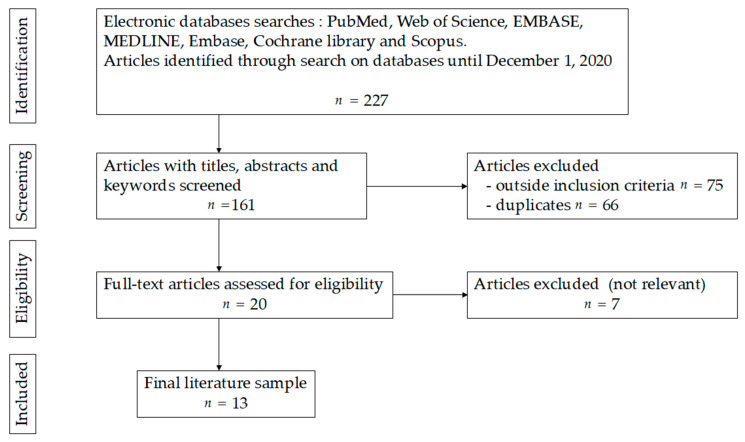
Study flow diagram for systematic review of the literature on liver defatting.

**Figure 2 jcm-10-01253-f002:**
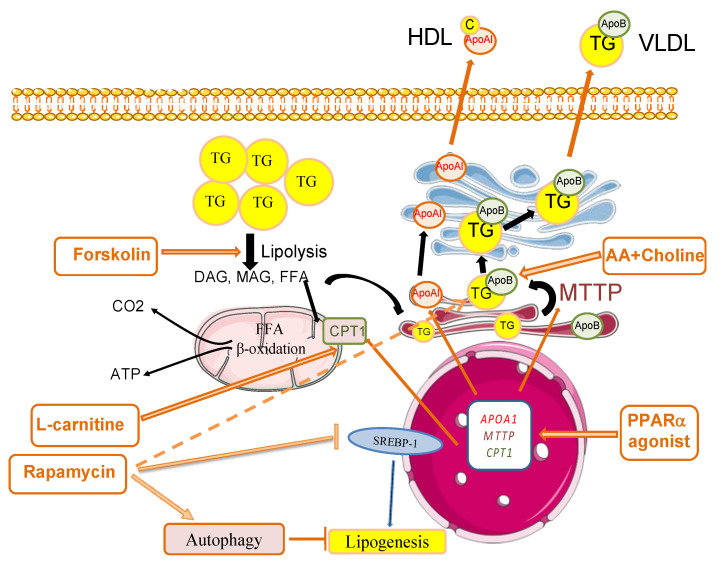
Main pathways and molecules used for liver defatting. Forskolin increases the phosphorylation of perilipin 5, a cell surface component of lipid droplets, and it activates the lipolysis of triglycerides (TG), leading to the generation of diacylglycerol (DAG), monoacylglycerol (MAG), glycerol and free fatty acids (FFAs). FFAs serve as substrates for both β-oxidation in mitochondria, and TG synthesis targeted to very low-density lipoprotein (VLDL) production in the endoplasmic reticulum. The activation of β-oxidation is triggered by supplementation with L-carnitine, a substrate of carnitine palmitoyltransferase 1 (CPT1) for the entry of FFAs into mitochondria. The addition of amino acids (AAs) and choline is aimed to foster VLDL synthesis. PPARα (peroxisome proliferator-activated receptor α) agonists induce the expression of PPARα-target genes including CPT1, microsomal triglyceride transfer protein (MTTP) and apolipoprotein A1 (APOA1), which contribute to FFA β-oxidation, VLDL production and cholesterol (C) export via high-density lipoprotein (HDL) formation, respectively. MTTP, a key enzyme in VLDL production, catalyzes the transfer of triglycerides to apolipoprotein B (ApoB) in the endoplasmic reticulum. Rapamycin has a number of effects that concur to decrease steatosis including the inhibition of mTOR, which promotes lipogenesis, the induction of triglyceride secretion and of macro-autophagy, a process in which cells engulf and degrade intracellular components including lipid droplets. This process maintains organelle quality control, acts as a survival mechanism and contributes to lipid droplet removal.

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
