# Peer review of "Ex-Vivo Pharmacological Defatting of the Liver: A Review"

_jcm, 2021, doi:10.3390/jcm10061253_

Round 1
Reviewer 1 Report
The paper offers a concise but precise overview of the major contributions on the field of defatting treatments for steatotic livers.
The methods are adequate for a systematic review, however there is a inconsistecy that needs to be addressed. The flow diagram in Figure 1 shows 13 articles finally selected for the review. In tables 1 and 2 only 12 articles are shown. Can you correct this discordance?
The sections of the review are well and clearly distinct in a short and focused summary of steatosis physiopathology, and in the analysis of in vitro / ex vivo and human / animal models found in the literature search. A final paragraph on future perspective offered by normothermic machine perfusion closes the discussion.
Other then some minor grammar corrections, I don't have other observations to raise.
Author Response
We thank reviewer N.1 for his kind comments on our review. We apologize for the mistake in Figure 1 and have modified the revised manuscript and Figures accordingly.
Reviewer 2 Report
NMP technology defatting liver steatosis is very interesting!
Elegant figure 2 and clear "summary" tables. Congratulations!
Author Response
We would like to thank Reviewer 2 for his kind comments. We also find liver defatting very interesting !
Reviewer 3 Report
Thank you for the invitation to review the manuscript entitled "Ex-vivo pharmacological defatting of the liver: a review" and submitted to the Journal of Clinical Medicine.
Authors perform an extensive review of a very novelty and interesting topic. In my opinion, the manuscript could be accepted for publication in its actual version.
Author Response
We thank reviewer n.4 for his kind comments.
Reviewer 4 Report
This systematic review has been conducted according to PRISMA guidelines and is based on a final literature sample of 13 full text articles out of 227 papers.
It makes a comprehensive synthesis on existing pharmacological therapies for liver defatting, with special focus on normothermic liver machine perfusion as an experimental ex-vivo translational model.
In surgical practice macro-steatosis over 60% is associated with liver damage and dysfunction and as a consequence those livers are usually declined for transplantation. The damage caused by steatosis comes from different mechanisms which finally lead to a reduced capacity to endure ischemia. These mechanisms are discussed in relation to main pathways and molecules for liver defatting drugs and cocktails thereof. Results of several in vitro and in vivo models are reviewed. Special focus is on the role of normothermic machine perfusion (NMP).
Major comments.
- This review is badly structured. Data from different researchers are mislocated in the proposed chapters:
Some examples are: Chapter 3.2.2 deals with “animal NMP models”. However, important experiments are mentioned of in vitro studies on isolated hepatocyte cultures: lines 137-140, lines 159-163 and lines 164-169. In addition, results of Raigani (lines 170-177) have been obtained in normothermic hyperoxic MP. Furthermore, under 3.2.3 “Human in vitro models”, experiments are described in HFD-fed mice and AML-12 mouse hepatocytes (lines 207-215), and in a porcine model of NMP (lines 240-243).
- Table 2 shows drugs used in experimental models for ex vivo defatting of steatotic livers by experimental (s)NMP: of six studies only 2 concern human livers and the total number of livers is only 7. This makes the conclusion at page 9 lines 300-302 “NMP is a powerful platform for assessing novel therapeutics on fatty livers, with the goal of reducing fat content and in the end optimizing fatty liver grafts for transplant” very preliminary and speculative.
- The proof of the pudding is in the eating and what is really missing are experimental data that defatting leads to improve liver function and ultimately better TX success. Studies are missing that liver defatting by drugs under NMP results in improved liver function as lactate and ammonia elimination, pH stabilization, bile production, gluconeogenesis, protein synthesis and mixed function oxidase. Fat reduction should lead to better function and survival of the graft. A critical analysis of this relevant aspect is not given by the authors.
Author Response
We thank reviewer n.4 for its insightful and relevant review. Here is a point by point response to his comments.
1- We apologize for the inconsistencies spotted in our review. The manuscript has been carefully and comprehensively revised accordingly.
2- We agree that our conclusion sentence may sound a little emphatic given the small number of human livers perfused yet in normothermia with the defatting cocktail. We modified this sentence (page 9 line 451) in the revised version of the manuscript accordingly : "In conclusion, NMP is a promising platform for assessing novel therapeutics on fatty livers, with the goal of reducing fat content and in the end optimizing fatty liver grafts for transplant. Novel pharmacological defatting therapies already demonstrated encouraging preliminary results which require further validation."
3- We totally agree with this relevant comment ; defatting should prove beneficial by improving the perfused liver function on biological parameters. We added the following critical sentence to the paragraph 3.3 "Defatting and ex-vivo human liver machine perfusion" in the revised version of the manuscript page 8, line 418: "Moreover, the effect of fat content reduction on liver function improvement still remains to be demonstrated through larger experimental studies focusing on surrogates of liver metabolism and transplantability criteria".
Round 2
Reviewer 4 Report
The manuscript has been sufficiently revised.